# Development of Amino Acids Functionalized SBA-15 for the Improvement of Protein Adsorption

**DOI:** 10.3390/molecules26196085

**Published:** 2021-10-08

**Authors:** Raquel Gutiérrez-Climente, Margaux Clavié, Jérémie Gouyon, Giang Ngo, Yoann Ladner, Pascal Etienne, Pascal Dumy, Pierre Martineau, Martine Pugnière, Catherine Perrin, Gilles Subra, Ahmad Mehdi

**Affiliations:** 1ICGM, University Montpellier, CNRS, ENSCM, 34000 Montpellier, France; 2IBMM, University Montpellier, CNRS, ENSCM, 34000 Montpellier, France; margaux.clavie@umontpellier.fr (M.C.); jeremie.gouyon@univ-lorraine.fr (J.G.); yoann.ladner@umontpellier.fr (Y.L.); Pascal.Dumy@enscm.fr (P.D.); catherine.perrin@umontpellier.fr (C.P.); 3IRCM, University Montpellier, Inserm, ICM, 34000 Montpellier, France; thi-hong-giang.ngo@inserm.fr (G.N.); pierre.martineau@inserm.fr (P.M.); martine.pugniere@inserm.fr (M.P.); 4I2C, University Montpellier, CNRS, ENSCM, 34000 Montpellier, France; pascal.etienne@umontpellier.fr

**Keywords:** sol–gel, silylated amino acids, SBA-15, protein adsorption, lysozyme

## Abstract

Ordered mesoporous materials and their modification with multiple functional groups are of wide scientific interest for many applications involving interaction with biological systems and biomolecules (e.g., catalysis, separation, sensor design, nano-science or drug delivery). In particular, the immobilization of enzymes onto solid supports is highly attractive for industry and synthetic chemistry, as it allows the development of stable and cheap biocatalysts. In this context, we developed novel silylated amino acid derivatives (Si-AA-NH_2_) that have been immobilized onto SBA-15 materials in biocompatible conditions avoiding the use of toxic catalyst, solvents or reagents. The resulting amino acid-functionalized materials (SBA-15@AA) were characterized by XRD, TGA, EA, Zeta potential, nitrogen sorption and FT-IR. Differences of the physical properties (e.g., charges) were observed while the structural ones remained unchanged. The adsorption of the enzyme lysozyme (Lyz) onto the resulting functionalized SBA-15@AA materials was evaluated at different pHs. The presence of different functional groups compared with bare SBA-15 showed better adsorption results, for example, 79.6 nmol of Lyz adsorbed per m^2^ of SBA-15@Tyr compared with the 44.9 nmol/m^2^ of the bare SBA-15.

## 1. Introduction

Highly ordered mesoporous siliceous materials have been intensively investigated as supports for enzyme or protein immobilization and have also found applications in bioseparation, enzyme catalysis, biosensors, and drug delivery [1,2,3,4,5]. Among them, Santa Barbara Amorphous material (SBA-15) [6,7] is the most widely used. SBA-15 is synthesized in the presence of Pluronic P123, a neutral triblock templating agent (tri-block copolymer poly(ethylene oxide)-poly(propylene oxide)-poly(ethylene oxide, PEO_20_PPO_70_PEO_20_). Compared to other hexagonal mesostructured materials such as the MCM-41 [7], it presents a wider pore size distribution (5–30 nm) and thicker walls. Those properties give SBA-15 a high surface area and an improved thermal, mechanical and chemical resistance [8]. The hydrothermal treatment required for its preparation is carried out at high temperatures (e.g., 130 °C) [9], thus reducing the microporosity in favor of macroporosity, and therefore stabilizing the structure of the SBA-15 compared to others.

Organic–inorganic hybrid macroporous materials can be prepared by the co-condensation method (direct synthesis) or by the grafting method (post-functionalization). Post-functionalization allows the incorporation of organic functional groups onto the pores of the material without altering their initial structural conformation. Despite its simplicity, the success of this approach is affected by several factors, such as the number and accessibility of residual silanols groups (SiOH) [10,11]. Therefore, to improve the yield of the reaction, this grafting step is preferentially carried out in anhydrous apolar solvents such as toluene at high temperatures. Indeed, in the presence of water or polar solvents, both the hydrolysis and condensation may occur also in solution, leading to the formation of large polysilanes that are not linked to the surface of the silica favored by the formation of intermolecular hydrogen bonds with the polar solvent or water. Conversely, when using anhydrous organic solvents, hydrolysis and subsequent functionalization can only occur near the silica surface, where the few water molecules are adsorbed [12,13]. Although some efforts have been made rearding the use of greener solvents such as ethanol instead of toluene [12,14], the post functionalization of such porous materials in water, in biocompatible conditions (i.e., PBS buffer) has not been explored so far.

This is of great interest for the use of such materials as catalytic adsorbents, where the presence of densely populated organic groups reduce the catalytic efficiency, probably due to the fact that the silanols present on the surface of the materials may act as weak acids, having a direct effect on the catalytic reactions [12,15]. It is also interesting in the case of the materials are used as drug delivery platforms, where degradation of the material is required. As already point out, silica degradation is favored when the material was prepared using lower temperatures (e.g., 60 °C) during the hydrothermal treatment [9]. However, it is also influenced by the incorporated organic groups [16]. For instance, Ratirotjanakul et al. [17] evaluated the effect of three amino acids (glycine, aspartic acid and cysteine) in mesoporous silica nanoparticles degradation. They observed that the presence of nucleophilic side chain (i.e., aspartic acid) induced a faster degradation of the silica nanoparticles while assisting the diffusion of nucleophiles to hydrolysis sites.

Besides promoting degradation, AA are very attractive as organic moieties compared to simple organic function. For instance, AA can act as stereo inducers. For example, Kuschel et al. [18] designed materials with chiral properties by using alanine and histidine as bridging organic group of silsesquioxane precursors for the synthesis of periodically ordered mesoporous organosilica materials (PMOs). AAs are also valuable ligands for metal complexation. Luechinger, M et al. [19] developed a biomimetic catalyst by immobilizing the histidine and glutamic acid onto the surface of M41S silica materials, which complexed with Fe(II) in aqueous solution. In addition, the variety of their side chains (polar, hydrophobic, positively or negatively charged), can be used to obtain surfaces which may improve cell adhesion [20] or favor [21]/avoid [22] protein adsorption. In our group, we consider that the grafting of amino acids on a material surface is a way to mimic a protein surface, and may favor interaction with other relevant proteins. This can be applied for the development of protein-imprinted materials by the inorganic approach, where the necessity of new sol–gel precursors has been highlighted [23].

In this context, we propose the synthesis of AA-modified mesoporous materials to use them as support for the adsorption of proteins. In contrast to the existing multi-step protocols (i.e., functionalization with 3-aminopropyl trimethoxysilane followed by the activation of AA and amide bond formation), our method proceeds in a single step without the need for organic reagents and toxic solvents (e.g., toluene).

First, we set up a protocol for the synthesis of hybrid silylated amino acids (AA) which can be adapted to the silylation of all types of amino amides (polar, hydrophobic, acidic, or basic). Interestingly, we also describe an alternative to silylate amino amides in aqueous media, which is highly interesting when the selected amino amide is insoluble in anhydrous organic solvents such as dimethylformamide, typically used in silylation protocols [17,24].

Then, we propose a protocol to functionalize SBA-15 supports with these hybrid precursors in soft conditions (water, neutral pH), and we evaluate their impact on the texture and stability of materials. Finally, we perform adsorption assays of a protein model (lysozyme), demonstrating the interest of the modification of bare SBA-15 (SBA-15@AA) for the control of protein adsorption.

## 2. Results and discussion

### 2.1. Synthesis of the Amino Acid Hybrid Blocks (Si-AA-NH_2_)

We prepared a set of AA with different types of sides chains, in order to mimic the diversity of protein surfaces, in the context of a protein sequence where both the N and C termini of each residue is engaged in amide bonds. In other words, the AA building blocks did not bear any charged residue on their C or N termini, restraining this role to the different side chains. Thus, the carboxylic acid was converted to primary amide, and the α-amino group was reacted with 3-isocyanatopropyltriethoxysilane (ICPTES) to yield urea function. Both functions could afford hydrogen bonds donors and acceptors, in a similar way to residues involved in a peptide chain.

Silylated amino amides (Si-AA-NH_2_) were prepared from corresponding commercial amino amides (H-AA-NH_2_) in dry DMF and basic media (Figure 1). In these conditions, and by using only 0.95 equivalent of ICPTES, the primary amine of AA reacted chemoselectively, avoiding unwanted reactions with side chain, in particular alcohol and phenol groups of Ser and Tyr, respectively. Moreover, this procedure limits the excess of ICPTES to react on the OMS material, leading to unwanted functionalization.

The starting amino amides are available as hydrochloride salts, and the non-nucleophilic base N,N-diisopropylethylamine (DIEA) was used to get the free amine in DMF. Solvent was evaporated under vacuum and the mixture was precipitated in cooled diethyl ether to remove DIEA and DMF traces. The final product was dried under vacuum. Unfortunately, DIEA proved to be difficult to remove except for silylated-tyrosinamide (Si-Tyr-NH_2_). Indeed, Si-Tyr-NH_2_ precipitated as a white powder in cooled diethyl ether, contrary to the other Si-AA-NH_2_, which precipitated as gel trapping DIEA (25 molar% for Si-Ser-NH_2_ and 50 to 60 molar% for Si-Arg-NH_2_ and Si-Leu-NH_2_). These results were confirmed by ^1^H NMR analyses (Figure 2).

To get rid of residual DIEA, the silylation was perform in heterogeneous media, using sodium bicarbonate as insoluble base in DMF. After completion of the reaction (24 h, stirring 400 rpm), sodium bicarbonate was removed by filtration and the solvent was evaporated under vacuum. Si-AA-NH_2_ were precipitated in cooled diethyl ether to remove DMF traces and obtained as powders without trace of any contaminants (Figure 3).

Unfortunately, this protocol cannot be used for AA poorly soluble in anhydrous organic solvents such as glutamine. In addition, the use of ICPTES requires anhydrous conditions, not only to avoid isocyanate conversion to primary amine through carbamic acid formation, followed by decarboxylation; but also the premature hydrolysis of triethoxysilane moiety and their subsequent condensation.

It was thus necessary to use a tailored silylating reagent, which could react in water-containing media on amino groups and bearing a silanol protecting group stable in water at basic pH. We prepared an imidazolecarboxamide silylating reagent from (3-aminopropyl)tris(trimethylsiloxy)silane and carbonyldiimidazole, in dry dichloromethane in the presence of DIEA. Once the reaction has been completed, the solvent was evaporated under vacuum, then silylating reagent **5** was precipitated in water and lyophilized. Imidazole carboxamide (**5**) proved to be soluble in water/acetonitrile mixture (35/65 *v*/*v*). It reacted with H-Gln-NH_2_ in this mixture with DIEA (5 eq) to yield Si-Glu-NH_2_
**6** (Figure 4) with high purity (see Appendix A for ^1^H and ^29^Si NMR analyses).

### 2.2. Synthesis and Functionalization of the SBA-15

SBA-15 materials were prepared by hydrolysis and polycondensation of TEOS under acid conditions (Figure 5, more details are given in Section 3.2.1) in the presence of Pluronic P123 as structure-directing agent.

Although classical functionalization of SBA-15 with toluene and reflux allows the covalent immobilization of organic functional groups in high concentrations, this approach was discarded after a first test due to the lack of solubility of all the Si-AA-NH_2_. Additionally, the harsh conditions and the use of toluene were not suitable for future applications. Thus, we investigated two types of procedures (Figure 5) to graft silylated amino acids in aqueous media: (i) water at pH 2 (adjusted with HCl 2M); and (ii) PBS (100 mM, pH 7) with a small volume of ethanol (4.45/1 *v*/*v*) and NH_4_F as catalyst. The employed conditions for each material as well as the denoted names are summarized in Table 1 (see Section 3.2.3 for more details).

#### 2.2.1. Texture and Morphology of the Mesoporous Material

The textural parameters such as specific surface area, pore size and pore volume were determined by measuring the N_2_ adsorption-desorption isotherms at −195.8 °C (Figure 6, Table 2). All samples (SBA-15 and SBA-15@AA) present a typical type IV isotherm and a hysteresis loop type H1, which is a typical characteristic of mesoporous materials. As expected, once the SBA-15 have been functionalized with the different Si-AA-NH_2_ (i.e., SBA-15@AA) all isotherms showed the inflection point shifts a lower partial pressure, indicating a decrease of the pore diameter (Table 2). We can observe that the pore volume after functionalization decreased only by 0.2–0.7 cm^3^/g and that the pore diameter decreased by less than 2 nm, confirming first that the textural parameters of the material were not significantly affected by the functionalization, and secondly that the functionalization did not block the pores. Of course, some slight differences in S_BET_ and consequently V_t_ and D_BJH_ can be observed depending on procedure and the Si-AA-NH_2_ grafted. Taking into consideration the reduction of surface area between the SBA-15 and SBA-15@AA (Figure 7), we can observe the effect of the procedure used (acid catalyst, pH 2, or nucleophilic catalyst, pH 7). At pH 2, the hydrolysis of the Si-AA-NH_2_ is favored while the condensation is reduced. On the other hand, at pH 7, the condensation is favored, and the hydrolysis occurs thanks to the F^−^. The greater the amount of Si-AA-NH_2_ grafted, the lower the surface area, and the volume and diameter of pores; it is not surprising that higher reductions of the surface area occur under neutral conditions.

Concerning the effect of the different Si-AA-NH_2_ used, the most significant differences were observed for the Si-Arg-NH_2_, probably due to the fact that it was the only Si-AA-NH_2_ positively charged whatever the pH value. At pH 2, the hydroxyl groups of the silica are protonated, which can create electrostatic repulsion with the positively charged Si-Arg-NH_2_, reducing the amount of Si-Arg-NH_2_ grafted onto the surface of the material compared with the grafting at pH 7. It is also worth noting that the percentages of surface loss are more homogeneous between all the Si-AA-NH_2_ when the grafting is carried out with nucleophilic catalysis at neutral pH, than with acid catalysis. Indeed, the surface area decreased by 20–36% compared to the nude SBA-15. This is of particular interest for when mixtures of different Si-AA-NH_2_ have to be grafted in similar ratio.

Density dropped between 0.38 and 0.79 cm^3^/g after Si-AA-NH_2_ grafting compared to the bare SBA-15. For hybrid materials obtained with Si-AA-NH_2_ which were not affected by the type of catalyst (i.e., Si-Tyr-NH_2_; Si-Ser-NH_2_), the density values remain constant. In contrast, a more significant difference (0.08 cm^3^/g) was observed for the materials obtained with Si-Arg-NH_2_ and Si-Leu-NH_2_, whose grafting was affected by the type of catalyst.

As presented in Figure 8, XDR patterns of all hybrid materials were the same as those obtained for original SBA-15. They are well defined and display three diffraction peaks characteristic of ordered hexagonal materials at 2θ = 0.884°, 1.53° and 1.77° corresponding to (100), (110) and (200) planes. In addition, the cell parameters of all materials (Table 2a) are around 12 nm, and the wall thickness varied from 1.6 to 3.1 nm, showing that the textural properties and the structure of materials are not affected by the grafting. The cell parameters, together with the pore diameter determined by the BJH (Barrett-Joyner-Halenda) method, were employed for the calculation of the wall thickness based on Equation (1) [9]:t = a − 0.95D_BJH_(1)

Only a small increase in wall thickness was expected after the grafting step. Indeed, we aimed to create a thin layer of amino acids, ideally a monolayer (see Section 2.2.2 for details). The average thickness increased by 1.12 nm or 0.8 nm for the pH 7 nucleophilic catalysis protocol or the acid-catalyzed protocol, respectively. Those results are in agreement with the other characterization results of the materials. For instance, the SBA-15@Leu pH 2, SBA-15@Leu pH 7 and SBA-15@Arg pH 7 possess the smallest pore sizes among synthesized materials, which was verified by N_2_ ads./des. isotherms. At the same time, the XRD patterns of selected samples showed the shift of reflex assign to (100) plane to the higher value of 2theta. This shift suggests the decrease in pore size diameter. Thus, the data given from N_2_ ads./des. isotherms are verified by XRD patterns.

Finally, hybrid materials were analyzed by FT-IR (Figure 9). As expected for such materials, the band of the Si-O-Si asymmetric stretching (st) vibrations at 1060 cm^−1^ with a shoulder at 1150 cm^−1^ was the most intense. This band, together with the bands around 3300 cm^−1^, indicate the presence of the silanol groups (-OH). The presence of the Si-AA-NH_2_ after the functionalization is confirmed by the presence of the bands between 2870–2990 cm^−1^ corresponding to the C-H st of the alkanes groups and by the amide bands at 1560 and 1660 cm^−1^, corresponding to the C=O st and overlapping N-H (bend) and C-N (st) in C-N-H, respectively. No difference between Si-AA-NH_2_ can be plotted, excepted for the tyrosine derivative, which presents a characteristic band at 1515 cm^−1^ assigned to the C-C st of the aromatic ring.

It is worth mentioning that those Si-AA-NH_2_ that are highly soluble in water (Si-Ser-NH_2_ and Si-Arg-NH_2_) present a greater difference between the two grafting protocols, while those that are not immediately soluble in water and became soluble during their hydrolysis when acid catalyst is used, or those which need to be solubilized first in EtOH for the protocol catalyzed by the nucleophilic catalyst, present similar results between the two methods. This means that, as expected, the solubility of the Si-AA-NH_2_ plays an important role in the grafting process. Therefore, the protocol at pH 7 is undoubtedly a better choice when a mixture of Si-AA-NH_2_ has to be used. Indeed, all hybrid amino acids may solubilize faster in water (or water/ethanol), meaning that the grafting reaction may start almost simultaneously for all of them. According to those results, we evaluated the protein adsorption on hybrid SBA-15@AA synthetized at pH 7.

#### 2.2.2. Quantification of the AA Grafted onto SBA-15

The amount of silanol groups in SBA-15 mesoporous silica is around 4–4.06 OH/nm^2^ [11,25]. Considering that each Si-AA-NH_2_ may share covalent bonds with 3 OH groups, the maximal charge of Si-AA-NH_2_ should be 1.33–1.53 molecules/nm^2^ for a monolayer, which corresponds to 2.545–2.21 × 10^−24^ mol Si-AA-NH_2_/nm^2^. SBA-15 N_2_ adsorption isotherms indicated a surface area of (466.9 m^2^/g, Table 2). On this basis, for the grafting we used 1–1.2 mmol of Si-AA-NH_2_ per g of SBA-15.

The amount of the AA actually grafted onto SBA-15 was determined by elementary analysis (EA) and thermogravimetric analysis (TGA). As disclosed in Table 3, similar results were obtained with both techniques. Figure 10 presents the TGA curves of the mass loss of the samples between 25 and 1000 °C. A small loss of mass (1.6%) at 200 °C is observed for bare SBA-15 that can be assigned to water evaporation and silica surface dehydration from to the silanol groups. Consequently, we considered only the mass loss between 200 and 900 °C to estimate the Si-AA-NH_2_ grafted. It is worth mentioning that at those temperatures it is still possible to observe the condensation of OH groups on to surface in addition to the removal of the organic groups, that may explain the differences observed between the both techniques.

The density of AA grafting onto SBA was expressed as number of molecules of Si-AA-NH_2_ per nm^2^ (Table 3). For example, a density of 1.16 Si-Tyr-NH_2_ per nm^2^ was calculated for SBA-15@Tyr pH 2 from the 18.36 mass loss. Except for Si-Glu-NH_2_, all the Si-AA-NH_2_ functionalized with a nucleophilic catalyst (SBA-15@AA pH 7) displayed a density ranging from 1 to 1.42 Si-AA-NH_2_ per nm^2^, which confirmed the creation of a monolayer of AA. This observation is in agreement with the values of hybrid layer thickness obtained from N_2_ adsorption isotherms.

#### 2.2.3. Characterization of the Isoelectric Point and Charges of the SBA-15@AA

The rationnale for grafting of amino acid residues on SBA-15 surface was to yield a biomimetic surface, which could recapitulate some feature of protein’s surface, thus favoring interaction with another protein of interest. The first type of non-covalent interaction we can expect is hydrogen bond acceptor or donor, through the CONH moieties flanking the amino acid. However, the main contributor of protein adsorption on the grafted surface should be ionic bonds, which may vary depending on the side chains, neutral, positively or negatively charged. In this context, and to envision which type of protein could be adsorbed easily, the zeta potential of each hybrid materials had to be measured as a function of pH. The results are represented in Figure 11.

As expected, at pH 2, the zeta potential of all material was positive, due to the protonation of the silanol groups (Si(OH_2_)^+^). When the hybrid materials were grafted with neutral or anionic amino acids, (i.e., Si-Tyr-NH_2_, Si-Ser-NH_2_, Si-Leu-NH_2_, Si-Glu-NH_2_) the zeta potential at pH 2 ranged between 4.5 and 10.3 mV, compared to 17.3 mV for the bare SBA-15. This may indicate that silanol groups are still present on the surface of the mesoporous material, despite the grafting of AA. The effect of silanol protonation (pKa 4–4.8) is still visible for SBA-15 at pH 4 (−2.02 mV) but not anymore for SBA-15@Tyr-NH_2_, SBA-15@Ser-NH_2,_ SBA-15@Glu-NH_2_ and SBA-15@Leu-NH_2_ (−16.3; −15.2; −10.3; −12.9 mV respectively). Surprisingly, at pH 6, the value for the Si-Glu-NH_2_ (−15.6 mV, pKa 4.25) is quite different for the others hybrid materials grafted with neutral amino acids (SBA-15@Tyr-NH_2_, SBA-15@Ser-NH_2_ and SBA-15@Leu-NH) or the bare SBA-15 ranging from −26 to −30 mV.

However, as expected, SBA-15@Arg-NH_2_ showed a singular behavior, due to the cationic guanidine side chain (pKa 12.48). Indeed, while the point of zero charge (pzc) is around pH 10 for SBA-15@Arg-NH_2_, for the bare SBA-15 and the other samples it is found to be around pH 3.

The Zeta potential measurements may afford some indications regarding the protein adsorption potential depending on the pH and the nature of the protein (pI, amino acid exposed to the surface, etc.). However, the multiple interactions that may occur with a protein due to the variety of AA onto the surface make very difficult to predict the behavior that a protein will have with those materials, a clear example of this complexity is detail in Section 2.4 for the case of Si-Arg-NH_2_.

### 2.3. Stability of the SBA-15@AA Materials

The hydrolysis of the siloxane bond between the organosilane and the silica which triggers the degradation of hybrid materials is a well-known phenomenon [26]. In this sense, we studied the stability of the SBA-15@AA materials in aqueous solutions. As the hydrolysis of siloxane bond is pH dependent, the study was performed not only in milli-Q water (pH 6.8), but also in water adjusted to pH 2 and pH 10. PBS (10 mM pH 7.5) was also evaluated, as it is the most common buffer used in bioapplications, as well as the solvent we used in further experiments (see Section 2.4). After one week, the amount of Si-AA-NH_2_ was quantified (Figure 12).

As shown in Figure 12, the loss of Si-AA-NH_2_ grafted onto the SBA-15 is lower than 5% except for the SBA-15@Arg. This confirms the stability of the hybrid materials functionalized with the novel Si-AA-NH_2_ presented in this work.

We hypothesize that the degradation of Arg-functionalized materials was due to a pentacoordinate silicon intermediate formation, resulting from the nucleophilic attack of the guanidine non-binding doublet, in a similar way to aminopropylsilsesquioxane [27]. The better stability of SBA-15@Arg in PBS can be explained by the formation of guanidium phosphate salts. The guanidine moiety was less prone to generate the pentacoordinate silicon intermediate, therefore reducing the degradation of the material.

### 2.4. Protein Adsorption onto the SBA-15@AA

Finally, adsorption of Lysozyme (Lyz) was studied on the different SBA-15@AA by batch equilibrium studies.

We chose lysozyme (Lyz) as the protein model due to its small size (14.4 kDa), which allows it to interact not only on the surface of the material, but also to penetrate into the pores. Additionally, the nature of the protein (isoelectric point (pI) 11) make this protein positively charged at pH < 11. Having a constant charge of the protein at different pHs simplifies the evaluation of the materials, as the electrostatic interactions between protein and Si-AA-NH_2_ remains constant at pH < 11.

First, the adsorption isotherms were determined for the SBA-15 at two pH values (acetic acid/acetate buffer, 10 mM, pH 3.6 and phosphate buffer saline 10 mM pH 7.5). As expected, the saturation point of the bare SBA-15 is highly dependent of the pH of the solution. At pH 3.6, the adsorption capacity of the SBA-15 for the lysozyme (Lyz) is 3.7 ± 0.8 nmol of Lyz per m^2^ of the material, while it was more than 10 times higher at pH 7.5 (39.7 ± 8.0 nmol Lyz/m^2^). This difference can be explained by the presence of silanol groups of the silica (around 4–4.06 OH/nm^2^ in the case of mesoporous silica materials [11,25]). At pH 7, they establish strong electrostatic interactions with the protein, while at pH 3.6, those interactions disappear (pKa = 4.8) [28].

Then, a concentration from the saturation part of the SBA-15 isotherm (350 µM) was selected for determining the amount of Lyz bound on the functionalized materials (SBA-15@AA). The results are summarized in Figure 13. Interestingly, in physiological conditions (pH 7.5), whatever the type of Si-AA-NH_2_ grafted and the global charge of the material (positively charged for Si-Arg-NH_2_), the adsorption capacity of the hybrid SBA-15 was significantly improved compared with SBA-15. Si-Tyr-NH_2_ grafting was the most efficient and improved the adsorption capacity of about 50%.

As already stated, the Zeta potential results suggested the presence of residual silanol (-OH) on the surface of the materials, which would also play a role in the adsorption of the model protein.

To get rid of the putative electrostatic interactions with the residuals silanol, the adsorption studies were also carried out at pH 3.6, where the zeta potential of SBA-15 was also null (Figure 11). As already mentioned, the adsorption capacity of the SBA-15 is significantly reduced at that pH as the OH are not negatively charged, and therefore they do not strongly interact with the positively charged protein. The Lyz adsorption was also significantly reduced for the hybrid materials (Figure 13), but the effect of the functionalization by the amino acid was much clearer. Surprisingly, the adsorption capacity of the SBA-15@Glu was significantly higher than the other materials despite the fact that the side chain of glutamic acid is not anymore in its anionic carboxylate form (pKa 4.25). The significant differences in adsorption by changing the pH is of high interest in order fields, for instance, for the development of electrophoresis capillaries, which require a low adsorption of protein at lower pH (Gouyon et al., unpublished work).

## 3. Materials and Methods

### 3.1. Reagents

Tetraethylorthosilicate (TEOS, ≥99.0%), ammonium fluoride (NH_4_F, ACS reagent, ≥98.0%), sodium chloride (NaCl, BioXtra, ≥99.5%), sodium phosphate dibasic (Na_2_HPO_4_, ≥98.5%), sodium phosphate monobasic (NaH_2_PO_4_, ≥99.0%), sodium hydroxide (NaOH, ≥98.0%), hydrochloric acid (HCl, ACS Reagent, ≥37%), phosphoric acid (H_3_PO_4_, ACS Reagent, ≥85%), dry dimethylformamide (DMF, ACS Reagent, ≥99.8%); %); dichloromethane (DCM, extra dry over molecular sieve, 99.8%); diethyl ether (Et_2_O, 99.8%) and Pluronic P123 (tri-block copolymer poly(ethylene oxide)-poly(propylene oxide)-poly(ethylene oxide, PEO_20_PPO_70_PEO_20_) were purchased from Sigma-Aldrich (St. Louis, MO, USA). Ethanol (EtOH, Normapur, 96%) was obtained from VWR Chemicals (Fontenay-sous-Bois, France). Acetonitrile (ACN, Chromasolv, ≥99.9%) was purchased from Honeywell Lab (Muskegon, MI, USA). L-argininamide dihydrochloride (H-L-Arg-NH_2_), L-isoglutamine (H-L-Glu-NH_2_, 95%), L-leucinamide hydrochloride (H-L-Leu-NH_2_, 98%), L-serinamide hydrochloride (H-L-Ser-NH_2_, 95%), L-tyrosinamide (H-L-Tyr-NH_2_, 95%), 1,1′-carbonyldiimidazole (CDI, 93%), 3-(triethoxysilyl)propyl isocyanate (ICPTES, 95%), 3-aminopropyl(trimethylsiloxy)silane (95%) and the N,N-diisopropylethylamine (DIEA) were purchased from ABCR (Karlsruhe, Germany).

Lysozyme from chicken egg white (Lys, lyophilized powder, ≥99.0%) was purchased from Sigma (St. Louis, MO, USA).

### 3.2. Synthesis

#### 3.2.1. Synthesis of the Mesoporous Silica Adsorbent (SBA-15)

SBA-15 support was prepared according to a published method [9]. Pluronic P123 triblock copolymer was used as neutral template agent and tetraethylorthosilicate (TEOS) as silica precursor. Briefly, 1 g of Pluronic P123 was solubilized in 5 mL of water and 10 mL of HCl 2 M. Once P123 was completely solubilized, 2.1 g of tetraethylorthosilicate was added (molar ratio of SiO_2_/P123/HCl/H_2_O, 1/0.017/2/80). Then mixture was maintained at 35 °C for 24 h (sol formation) and 2 days at 130 °C (without stirring) in a Teflon-lined autoclave (hydrothermal treatment). Materials were then filtered, washed with water (until complete neutralization) and ethanol. After 24 h drying at 80 °C, samples were calcined at 550 °C.

#### 3.2.2. Synthesis of the Silylated Amino Amides

An amount of 1 eq of the H-AA-NH_2_ was dissolved in 4 mL of dry dimethylformamide. An amount of 3 eq of the base N,N-diisopropylethylamine (DIEA) or sodium bicarbonate (NaHCO₃) was added followed by the addition of 0.95 eq of 3-isocyanatopropyl)triethoxysilane (ICPTES). The mixture was stirred at room temperature under argon until the reaction was completed (see Table 4 for further details). The solvent was evaporated under vacuum and the resulting oil was precipitated in cooled diethyl ether before drying under vacuum.

To prepare Si-Glu-NH_2_, (3-imidazolecarboxamide) tris(trimethylsiloxy)silane was first synthesized according to the following protocol: (3-aminopropyl) tris(trimethylsiloxy)silane (1 eq, 5 mmol, 2 mL) was dissolved in dry DCM (8 mL) with DIEA (0.6 eq). CDI (1.1 eq) was dissolved in dry DCM (30 mL) and added onto the silane solution. The mixture was stirred at room temperature under argon for three days. The solvent was evaporated under vacuum and the resulting oil was precipitated in water before being lyophilized (93% yield). Then, isoglutamine (Si-Glu-NH_2_, 1 eq, 0.86 mmol, 126 mg) was dissolved in a mixture of water/acetonitrile (1/1 *v*/*v*) and DIEA (5 eq) was added. (3-imidazolecarboxamide)tris(trimethylsiloxy)silane (1.1 eq) was dissolved in a mixture of water/acetonitrile (1/2.5 *v*/*v*) and was added to the isoglutamine solution. The mixture was stirred at 50 °C for three hours before being lyophilized. The obtained crude product was washed with diethyl ether to remove residual DIEA and was dried under vacuum (52% yield).

#### 3.2.3. Functionalization of the SBA-15

Two protocols were used for the functionalization of the SBA-15 in order to evaluate the effect of using H^+^ or F^−^ as catalyst for the reaction. For the first one, where the reaction was catalyzed by H^+^, 1 g of SBA-15 was incubated with 1mmol of Si-AA-NH_2_ in 22 mL of water pH 2. Reaction was left at room temperature under stirring (250 rpm) for 24 h. Longer time (48 h) had not a significant effect on the amount of Si-AA-NH_2_ grafted.

In the second protocol, 1 g of SBA-15 was incubated with 1 mmol of Si-AA-NH_2_ in 22 mL of a mixture (1/4.45 *v*/*v*) of ethanol and phosphate buffer (100 mM, pH 7). 250 mg of NH_4_F was added as catalyst and reaction was left under stirring (250 rpm), at room temperature for 16 h.

Once the grafting had been carried out, material was washed by centrifugation: three times with a mixture of water/ethanol (2/1 *v*/*v*), once with water and once with ethanol. The resulting hybrid material was dried under vacuum at 30 °C.

The SBA-15 without functionalization (bare SBA-15) was employed as control. The obtained materials were denoted as shown in Table 1. Five different Si-AA-NH_2_ were employed in this study: Si-Tyr-NH_2_, Si-Ser-NH_2_, Si-Arg-NH_2_, Si-Leu-NH_2_, Si-Glu-NH_2._

### 3.3. Characterization

^1^H and ^29^Si DEPT NMR spectra were recorded at 25 °C in deuterated dimethyl sulfoxide (DMSO-d6) on a Bruker Avance III 500 MHz spectrometer equipped with Helium BBO Cryoprobe operating at 500 MHz and 99.4 MHz, respectively. Chemical shifts (δ) are reported in parts per million (ppm) using residual non-deuterated solvents as internal references. Coupling constants are measured in Hertz.

The functionalization of the SBA-15 was confirmed by FT-IR, thermogravimetric analysis (TGA) elementary analysis (EA). Infrared spectra were recorded with a 4 cm^−1^ resolution on a PerkinElmer Spectrum Two FT-IR spectrometer equipped with a diamond crystal attenuated total reflectance (ATR) unit. The quantification of the Si-AA-NH_2_ grafted was determined by EA (%C, %H, %N), which was performed in a Elementar Vario Micro Cube. TGA was conducted using a NETZSCH STA 409 PC/PG. The samples were heated in a Teflon crucible at a heating rate of 10 °C/min.

X-ray diffraction (XRD) was used to identify the crystal structure of SBA-15 samples and determine if that structure has been affected by the functionalization. For that, an X’pert Pro diffractometer (PanAnalytical) was used to record the X-ray diffractograms with Ni-filtered Cu radiation (λ<α> = 1.5418 Å). The incident beam was conditioned using two fixed slits with 1/32 and 1/16 rad opening angles. The diffracted intensity was detected using an ultrafast X’celerator detector. The material was deposited on a plexiglass sample holder and measurements were performed at room temperature (25 °C) between 2θ = 0.616° and 7.000° using rapid scans (analysis time = 10 min).

The effect of the functionalization on the porosity and surface was also characterized. The nitrogen adsorption–desorption isotherms at −195.8 °C of the silica were recorded on a Micromeritics Gemini III 2375 apparatus. Previous measurement, all samples were degassed overnight at 80 °C until a stable static vacuum of 0.1 mbar was reached. The specific surface area was determined using the BET equation. The pore size distribution was calculated using the BJH method, and the microporous volume was estimated by the t-plot method using the Harkins and Jura standard isotherm. Surface charges were measured with Zetasizer Nano equipment (Nano-S) from Malvern Panalytical (France) at a concentration of 240 µg/mL. The measurement parameters were as follows: a laser wavelength of 633 nm, a scattering angle of 173°, a measurement temperature of 25 °C, a dispersant refractive index of 1.33 and material refractive index of 1.45. The pHs of the aqueous solutions were previously set with the addition of 2 M HCl or 2 M NaOH solutions

Density was determined by the AccuPyc II 1340 pycnometer, which measures the pressure change of helium (99.995% pure) in a calibrated volume. A 1 cm^3^ sample cell was employed, and density was calculated by difference of the volume measured of the sample cell empty and once that the sample has been introduced.

### 3.4. Protein Adsorption Test

Traditional batch equilibration method was used for the quantification of the protein adsorption onto the bare SBA-15 and the SBA-15@AA materials. For that, 30 mg of the bare or SBA-15 or SBA-15@AA was incubated in presence of 2 mL of a protein solution at room temperature. Different concentrations of proteins in a range between 0 and 600 µM were employed. After 20 h of incubation, the suspension was centrifuged at 16,000 rpm for 10 min, and the supernatant was centrifuged to remove all silica present in the solution. The protein standards used for the study and for the calibration curve were also centrifuged under the same conditions to verify that protein was not precipitating. The solutions were measured by UV-VIS, all the spectra from 400–200 nm were determinate to detect possible degradation of the protein. The maximum wavelength (280 nm) was used for the calculations. Bounded amount of each protein adsorbed was calculated based on the following equation:(2)B=C0−Ceq∗VmSBET
where *B* (nmol/m^2^) corresponds to the amount of protein adsorbed per gram m^2^ of the material; *C* refers to the concentration of the protein added (*C*_0_) and founded in the supernatant after equilibrium (*C_eq_*) and *V* represents the volume of the protein solution (typically 2 mL). Finally, m is the mass of adsorbent added (typically 30 mg) and *S* (m^2^/g) indicates the surface area of the material obtained by BET equation. As the surface area of each material after functionalization is significantly different with respect to the bare SBA-15, we present the results in terms of m^2^ instead of g in order to have more comparable results.

In the case of the adsorption isotherms carried out for SBA-15, the obtained results were fitting as described by Bhatt et al. [29].

### 3.5. Degradation Studies of the Modified SBA-15-AA Materials

An amount of 30 mg of SBA-15@AA was dispersed in 2 mL of one of the following solutions: milli-Q water, milli-Q water pH 2, milli-Q water pH 10 or PBS (10 mM, pH 7). Milli-Q water was adjusted to pH 2 and pH 10 with HCl 2 M and NaOH 2 M, respectively. The samples were incubated at room temperature for a week. The samples were then centrifuged at 16,000 rpm for 10 min, washed twice with milli-Q water, recovered by centrifugation, and dried under vacuum at 30 °C overnight.

The amount of Si-AA-NH_2_ which remained grafted was determined by elemental analysis.

## 4. Conclusions

In this study, we disclosed the preparation of silylated amino acids that were isolated and conserved as powder, thus facilitating their use as convenient building blocks for further experiments. Interestingly, we developed a new silylating reagent which allowed the preparation of silylated organic molecules in water. Five hybrid amino acids were prepared and used to functionalize an SBA-15 in aqueous conditions, at pH 7 without affecting the textural and structural properties of the ordered mesoporous material.

Amino acid grafted hybrid materials were used as adsorbents for the protein Lyz, with an improvement of the adsorption capacity for all the SBA-15@AA compared to the bare SBA-15 at pH 7. Under acidic conditions, the adsorption capacity of SBA-15 was strongly reduced as well as for the SBA-15@AA with the exception of SBA-15@Glu.

The soft conditions of functionalization, combined to the simplicity of the synthesis amino acid silylated precursors and their diversity in terms of side chains, should give birth to interesting applications, in particular in bioseparation, biosensing, and drug delivery.

## Figures and Tables

**Figure 1 molecules-26-06085-f001:**
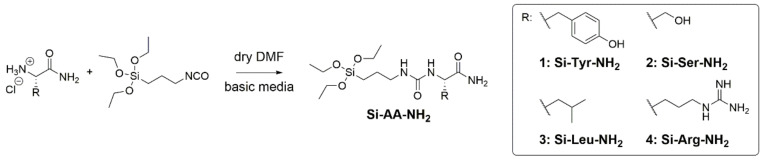
General way of amino amides silylation with ICPTES.

**Figure 2 molecules-26-06085-f002:**
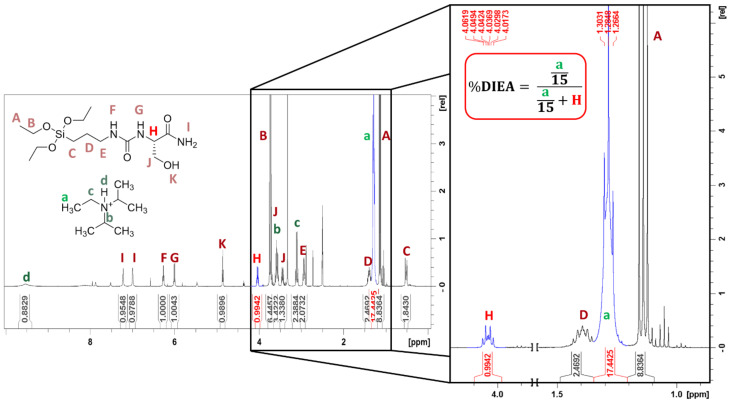
DIEA percentage calculation by ^1^H NMR (500 MHz, 25 °C). Example of Si-Ser-NH_2_. DIEA% was determined by using DIEA CH_3_ signals (1.26–1.31 ppm) and amino amide CHα (4.01–4.26 ppm depending on amino amide).

**Figure 3 molecules-26-06085-f003:**
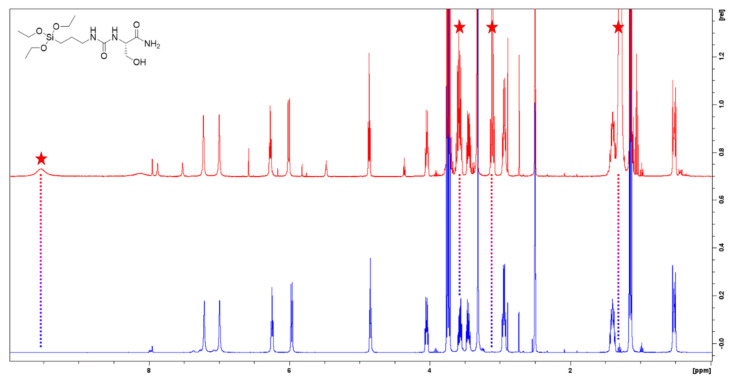
Comparison of ^1^H NMR spectra of Si-Ser-NH_2_ synthesized with DIEA (red) and sodium bicarbonate (blue).

**Figure 4 molecules-26-06085-f004:**
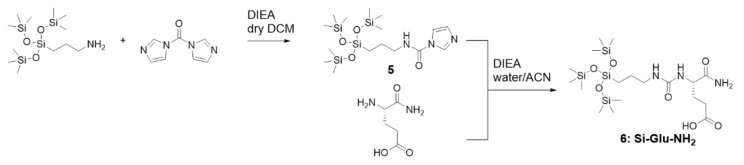
Silylation of isoglutamine with N-[3-tris(trimethylsiloxy)silylpropyl]-imidazole-1-carboxamide (**5**).

**Figure 5 molecules-26-06085-f005:**
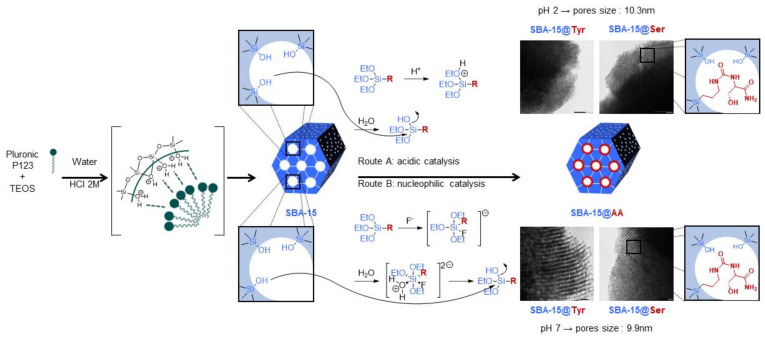
Schema of the SBA-15 functionalization with Si-AA-NH_2_.

**Figure 6 molecules-26-06085-f006:**
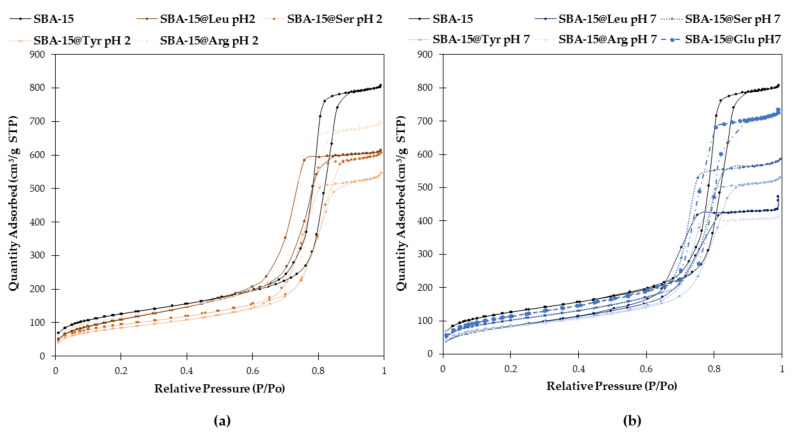
Nitrogen isotherms at 77K of bare SBA-15 (black) and functionalized SBA-15@AA at pH 2 (**a**) and pH 7 (**b**).

**Figure 7 molecules-26-06085-f007:**
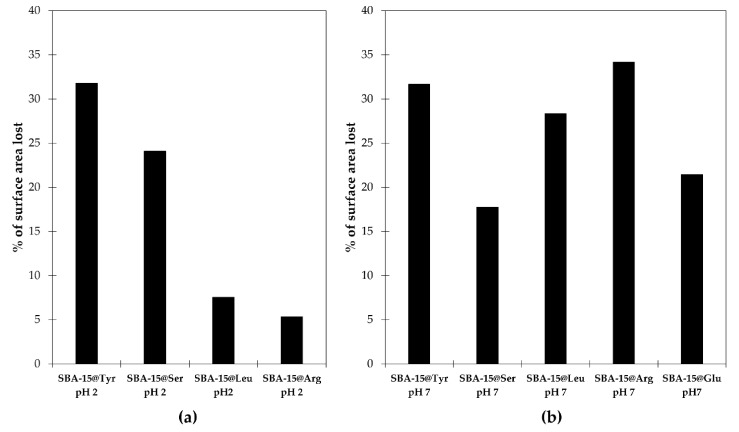
Percentage of surface lost for each material after the grafting step at pH 2 (**a**) and at pH 7 (**b**).

**Figure 8 molecules-26-06085-f008:**
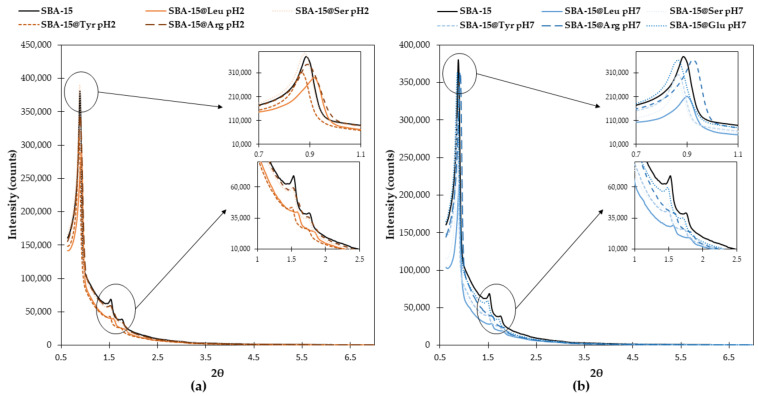
X-ray diffraction patterns of SBA-15 and functionalized SBA-15@AA obtained by acid catalysis (**a**) or nucleophilic catalysis at pH 7 (**b**).

**Figure 9 molecules-26-06085-f009:**
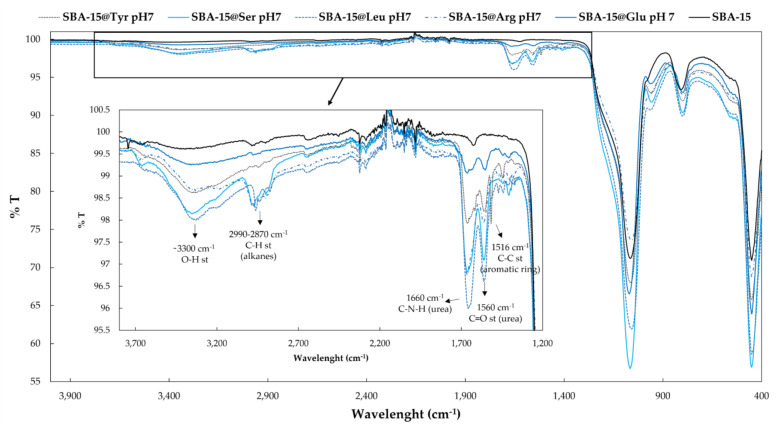
FT-IR spectra of the bare SBA-15 and the functionalized SBA-15@AA at pH 7.

**Figure 10 molecules-26-06085-f010:**
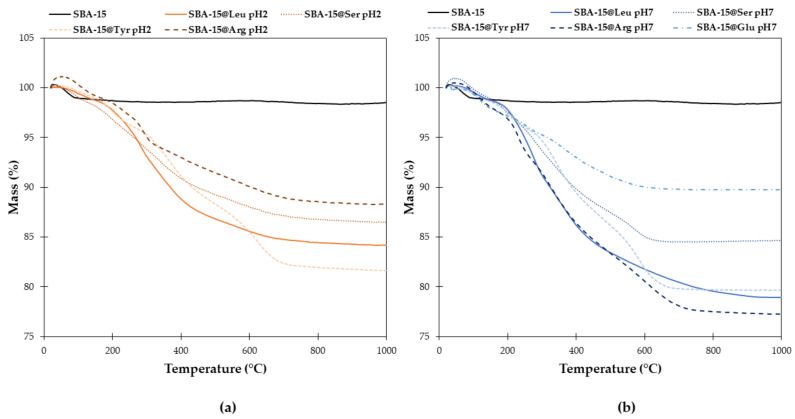
TGA curves of SBA-15 (black) and the SBA-15@AA at pH 2 (**a**) and pH 7 (**b**).

**Figure 11 molecules-26-06085-f011:**
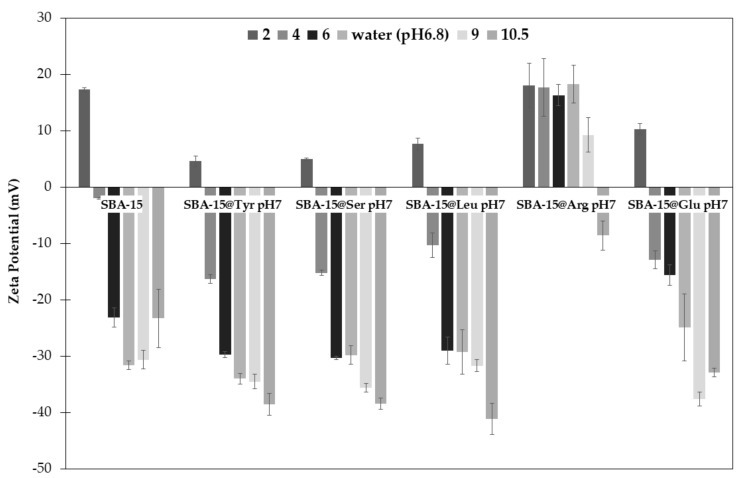
Zeta potential of bare SBA-15 and the SBA-15@AA samples as a function of pH. Measurements were performed at 25 °C and at a concentration of 0.24 mg/mL.

**Figure 12 molecules-26-06085-f012:**
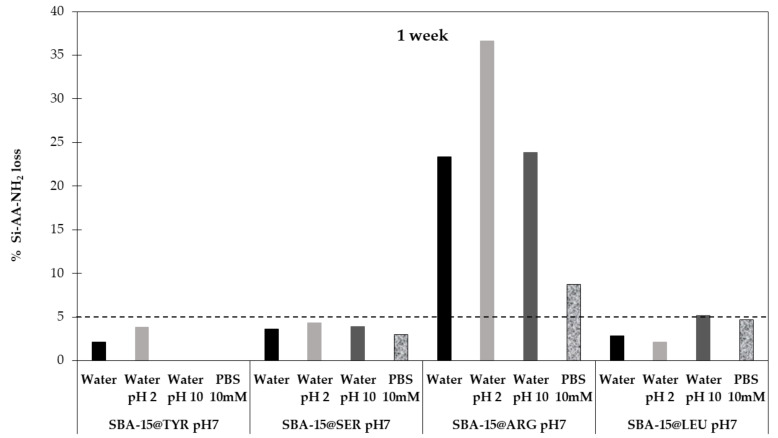
Percentage of Si-AA-NH_2_ grafted lost after 1 week of incubation at different aqueous conditions.

**Figure 13 molecules-26-06085-f013:**
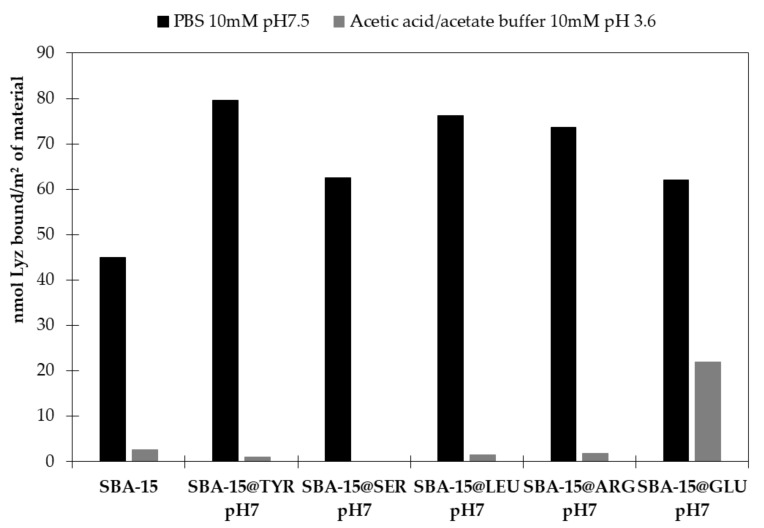
Amount of Lysozyme bound per m^2^ of material at physiological conditions, i.e., PBS 10 mM pH 7.5 (**black**) and under acid conditions, i.e., acetic acid/acetate 10 mM pH 3.6 (**grey**). (See Section 3.4 for details.)

**Table 1 molecules-26-06085-t001:** Summary of the functionalized materials, the functionalization conditions used and the denoted name.

Si-AA-NH_2_	Solvent Used	Catalyst	Sample Name	General Name
-	-	-	SBA-15	SBA-15
Si-Tyr-NH_2_	water	HCl	SBA-15@TYR pH 2	SBA-15@AA
Si-Tyr-NH_2_	PBS (100 mM pH 7)/ethanol (4.45/1 *v*/*v*)	NH_4_F	SBA-15@TYR pH 7	
Si-Ser-NH_2_	water	HCl	SBA-15@SER pH 2	
Si-Ser-NH_2_	PBS (100 mM pH 7)/ethanol (4.45/1 *v*/*v*)	NH_4_F	SBA-15@SER pH 7	
Si-Leu-NH_2_	water	HCl	SBA-15@LEU pH 2	
Si-Leu-NH_2_	PBS (100 mM pH 7)/ethanol (4.45/1 *v*/*v*)	NH_4_F	SBA-15@LEU pH 7	
Si-Arg-NH_2_	water	HCl	SBA-15@ARG pH 2	
Si-Arg-NH_2_	PBS (100 mM pH 7)/ethanol (4.45/1 *v*/*v*)	NH_4_F	SBA-15@ARG pH 7	
Si-Glu-NH_2_	PBS (100 mM pH 7)/ethanol (4.45/1 *v*/*v*)	NH_4_F	SBA-15@GLU pH 7	

**Table 2 molecules-26-06085-t002:** Properties of the SBA-15 and SBA-15@AA samples. Surface area (S_BET_), C-value (C_BET_), pore volume (V_t_) and diameter (D_BJH_) were obtained by N_2_ adsorption analysis; density was measured with a pycnometer and *d*-Spacing (d100), cell parameter (a) and wall thickness (t) were determined by XDR measurements.

	N_2_ Adsorption Analysis				Pycnometer	XDR		
Sample	S_BET_ (m^2^/g)	C_BET_	V_t_ (cm^3^/g)	D_BJH_ (nm)	Density (cm^3^/g)	d100 (nm)	a (nm)	t (nm)
SBA-15	466.9 ± 3.4	145.0	1.3	10.8	2.76 ± 0.09	10.3	11.9	1.6
SBA-15@TYR pH 2	302.3 ± 2.1	98.1	0.8	10.2	2.08 ± 0.01	10.5	12.1	2.5
SBA-15@TYR pH 7	302.8 ± 2.2	108.3	0.8	10.8	2.01 ± 0.06	10.5	12.1	1.9
SBA-15@SER pH 2	336.4 ± 2.4	113.7	0.9	11.2	2.22 ± 0.06	10.3	11.9	1.3
SBA-15@SER pH 7	364.4 ± 2.2	107.3	0.9	9.6	2.25 ± 0.08	10.5	12.1	3.0
SBA-15@LEU pH 2	406.2 ± 1.2	56.5	1.0	9.3	2.38 ± 0.01	10.0	11.5	2.6
SBA-15@LEU pH 7	314.9 ± 1.2	47.6	0.8	9.3	2.30 ± 0.17	10.1	11.7	2.9
SBA-15@ARG pH 2	416.0 ± 2.7	109.2	1.1	10.4	2.19 ± 0.00	10.3	11.9	2.2
SBA-15@ARG pH 7	289.2 ± 1.4	95.4	0.6	8.8	2.27 ± 0.00	10.0	11.5	3.1
SBA-15@GLU pH 7	410.6 ± 2.5	83.4	1.1	11.1	1.97 ± 0.01	10.5	12.1	1.6

**Table 3 molecules-26-06085-t003:** Quantification of the Si-AA-NH_2_ grafted onto the SBA-15 surface by EA and TGA.

	EA					TGA		
Sample	%C	%H	%N	mmol Si-AA-NH_2_ Grafted/g SBA-15	% Mass Loss (200–900 °C)		mmol Si-AA-NH_2_ Grafted/g SBA-15	Molecules of Si-AA-NH_2_/nm^2^
SBA-15	0.17 ± 0.07	0.49 ± 0.02	0.00 ± 0.00	0.00	0.31		0.00	
SBA-15@TYR pH 2	9.29 ± 0.13	1.64 ± 0.03	2.38 ± 0.05	0.69	18.36		0.50	1.16
SBA-15@TYR pH 7	10.99 ± 0.27	1.82 ± 0.06	2.92 ± 0.06	0.89	20.33		0.61	1.28
SBA-15@SER pH 2	4.60 ± 0.09	1.35 ± 0.03	2.09 ± 0.02	0.56	13.49		0.56	1.00
SBA-15@SER pH 7	7.19 ± 0.02	1.78 ± 0.00	3.38 ± 0.04	1.00	15.34		0.68	1.05
SBA-15@LEU pH 2	7.54 ± 0.07	1.80 ± 0.03	2.43 ± 0.01	0.68	15.81		0.55	1.14
SBA-15@LEU pH 7	10.76 ± 0.11	2.39 ± 0.03	3.64 ± 0.00	1.12	21.06		1.11	1.17
SBA-15@ARG pH 2	4.24 ± 0.16	1.46 ± 0.00	2.46 ± 0.09	0.32	11.68		0.36	0.55
SBA-15@ARG pH 7	8.47 ± 0.17	2.12 ± 0.08	5.61 ± 0.15	0.84	21.03		0.60	1.42
SBA-15@GLU pH 7	4.38 ± 0.12	1.61 ± 0.03	1.26 ± 0.07	0.33	10.16		0.52	0.53

**Table 4 molecules-26-06085-t004:** Summary of the synthesis conditions of the hybrid amino-acid-based precursors.

Si-AA-NH_2_	H-AA-NH_2_ (mg)	DMF (mL)	DIEA (mL)	NaHCO_3_ (mg)	ICPTES (mL)	Reaction Time (h)	Yield (%)
Si-Tyr-NH_2_	540	12	1.580	-	0.704	4	87
Si-Ser-NH_2_	141	4	-	420	0.235	24	84
Si-Arg-NH_2_	492	5.3	-	840	0.470	24	90
Si-Leu-NH_2_	253	4	-	0.630	0.353	24	83

## Data Availability

Not applicable.

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
