# Peer review of "Development of Amino Acids Functionalized SBA-15 for the Improvement of Protein Adsorption"

_molecules, 2021, doi:10.3390/molecules26196085_

Round 1

Reviewer 1 Report

The authors investigated the synthesis of various amino acid functionalized SBA-15. The synthesized catalysts were tested in adsorption of lysozyme at different pHs. The amino acid functionalized silicas showed better adsorption properties than bare SBA-15. The results are suitable for publication in Molecules journal. However, I think that manuscript should be improved before its publication. Some issues should be considered:

  1. Did the authors dry the samples before N2 adsorption/desorption measurements? If yes, at which temperature it was done? Did it influence on stability of organic species anchored on SBA-15? Please add the temperature of samples drying to the section 3.3.
  2. Table 2 and Figure 8: The SBA-15@LEU pH2, SBA-15@LEU pH7 and SBA-15@ARG pH7 possess the smallest pore sizes among synthesized materials, which was verified by N2 ads./des. isotherms. At the same time the XRD patterns of selected samples showed the shift of reflex assign to (100) plane to the higher value of 2theta. The such shift suggests the decrease in pore size diameter. Thus, the data given from N2 ads./des. isotherms are verified by XRD patterns. I kindly recommend to add this comment to the paper.
  3. Why the authors did not comment the better efficiency of Si-AA-NH2 grafting onto the SBA-15 at pH7 in comparison to pH2? What was the reason of such phenomenon?
  4. In my opinion it will be worth to add the DTG curve (together with TG) in figure 10. It will be helpful to define the ranges of organic phase decomposition. It gives the information to which temperature the catalyst is stable.
  5. Did the authors perform the reuse test of lysozyme adsorption for the most efficient catalyst in several reaction stages? I kindly recommend to add the such data to the manuscript.
  6. At which temperature the SBA-15 functionalization with Si-AA-NH2 was performed? Please add the required data to the section 3.2.3.
  7. Figure 7: please add the same maximum value in “y” axis for both (a) and (b) figures. It will be better visible the differences between both pHs.
  8. Please standardize the unit of temperature. The used units in the text are °C and K.
  9. The line 114: “hydrogel bonds” should be replaced by “hydrogen bonds”.
  10. The line 188: pore volume decreased of “0.2-0.7” not of “0.2-1” cm3/g.
  11. The line 280: “Table 2” should be replaced by “Table 3”.
  12. The line 291: “grating” should be replaced by “grafting”.

Author Response

Thank you for giving us the opportunity to submit a revised draft of our manuscript titled “Development of amino acids functionalized SBA-15 for the improvement of protein adsorption” to Molecules. We appreciate the time and effort that you have dedicated, as well as your valuable feedback on our manuscript. We are grateful for your insightful comments. We have been able to incorporate changes to reflect most of the suggestions provided. Please see below, in blue, for a point-by-point response to all the comments and concerns. All page numbers/lines refer to the revised manuscript file with tracked changes.

In addition to the above comments, all spelling and grammatical errors pointed out by the reviewers have been corrected. We have highlighted all the changes in the manuscript.

We look forward to hearing from you and responding to any further questions and comments you may have.

Sincerely,

Raquel Gutiérrez-Climente/Gilles Subra/Ahmad Mehdi

Reviewer 2 Report

The main advantage of this work is the synthesis of new hybrid precursors (silylated amino acids) that allow to modify the surface of mesoporous silica SBA-15 with amino acids (grafting) in aqueous  medium, avoiding  use of toxic solvents. The prepared materials were characterized using various physicochemical methods. It was shown that the surface modification contributed to an increase in the adsorption of lysozyme on the hybride materials due to electrostatic interactions.

I recommend this work for publication in the journal after minor revision.

The questions and notes:

  1. The quality of Fig.2 needs be improved. The Figure is unreadable.
  2. Page 8, line 248 and Fig.9. The synthesized hybrid materials contain amide groups. The bands at 1660 and 1560 cm-1 are attributed to the amide groups and called “amide” but not urea. They are assigned to C=O (str) and overlapping N-H (bend) and C-N (str) in C-N-H and not vise versa.   
  3. Page 10. line 278. One of the methods used for determination of the amount of AA grafted onto SBA-15 was TGA. The authors used the mass loss between 200 and 900°C to estimate the Si-AA-NH2 grafted. However, when the samples were  heated in this temperature range, condensation of surface OH groups occurs in addition to the removal of the organic groups. The obtained results have a low accuracy.
  4. All hybrid materials were synthesized at pH 2 and pH 7 using different catalysts (for example, SBA-15@TYR pH2  and SBA-15@TYR pH7). However, zeta potentials, stability and the protein adsorption were investigated only onto the samples prepared at pH 7. Why?  In some cases, the pore size and volumes for materials prepared at pH 2 are higher than those for materials prepared at pH 7 (Table 2).

Author Response

(The authors gave the same response as above.)
